# Optimization of cytotoxic activity of *Nocardia* sp culture broths using a design of experiments

**Alba Noël[1]¤, Gwendoline Van Soen[1], Isabelle Rouaud[1], Eric Hitti[2], Sophie Tomasi[1]\***

**1** Univ Rennes, CNRS, ISCR–UMR 6226, Rennes, France, **2** LTSI, UMR_S 1099, UFR Sciences Pharmaceutiques et Biologiques, Rennes, France

¤ Current address: I2BC, CNRS–UMR 9198, Orsay, France
\* sophie.tomasi@univ-rennes1.fr

**Data Availability Statement:** All relevant data are within the paper and its Supporting Information files.

**Funding:** The Funder "Ligue contre le Cancer 35" gave financial support (named BaLI) for the

## Abstract

In the context of research for new cytotoxic compounds, obtaining bioactive molecules from renewable sources remain a big challenge. Microorganisms and more specifically Actinobacteria from original sources are well known for their biotechnological potential and are hot-spots for the discovery of new bioactive compounds. The strain DP94 studied here had shown an interesting cytotoxic activity of its culture broth (HaCaT: $IC_{50}$ = 8.0 ± 1.5 μg/mL; B16: $IC_{50}$ = 4.6 ± 1.8 μg/mL), which could not been explained by the compounds isolated in a previous work. The increase of the cytotoxic activity of extracts was investigated, based on a Taguchi L9 orthogonal array design, after DP94 culture in TY medium using two different vessels (bioreactor or Erlenmeyer flasks). Various culture parameters such as temperature, pH and inoculum ratio (%) were studied. For experiments conducted in a bioreactor, stirring speed was included as an additional parameter. Significant differences in the cytotoxic activities of different extracts on B16 melanoma cancer cell lines, highlighted the influence of culture temperature on the production of cytotoxic compound(s) using a bioreactor. A culture in Erlenmeyer flasks was also performed and afforded an increase of the production of the active compounds. The best conditions for the highest cytotoxicity ($IC_{50}$ on B16: 6 ± 0.5 μg/mL) and the highest yield (202.0 mg/L) were identified as: pH 6, temperature 37°C and 5% inoculum.

## Introduction

Some antibiotics such as actinomycin, streptothricin and streptomycin have been discovered since the forties [1]. Actinobacteria also produce a wide variety of secondary metabolites with interesting biological activities such as insecticidal (e.g. avermectins from *S. avermitilis* [2]), antifungal (e.g. amphotericin B from *S. nodosus* [3]), antiviral (e.g. hygromycin from *S. hygroscopicus* [4]), and cytotoxic activities (e.g. bleomycin from *Streptomyces verticillus* [5] or asterobactin from *Nocardia asteroides* [6]). These bacteria are also well described for their interesting biotechnological applications [7]. Actinobacteria are widespread in environment making them

purchase of a Bioreactor BioFlo 115®. The funders had no role in study design, data collection and analysis, decision to publish, or preparation of the manuscript.

**Competing interests:** The authors have declared that no competing interests exist.

**Abbreviations:** OSMAC, One Strain Many Compounds; TY, Tryptone Yeast medium; LBm, modified Lysogeny Broth; MB, Marine Broth; ISP2, International Streptomyces Project 2; IC$_{50}$, half maximal inhibitory concentration; ANOVA, Analysis of Variance; DO, Dissolved Oxygen; DOE, Design of Experiments; DMSO, Dimethylsulfoxyde; NaCl, sodium chloride; MeOH, methanol; EtOAc, ethyl acetate; RE, resin extract; SE, supernatant extract; NaOH, sodium hydroxide; H$_2$SO$_4$, sulfuric acid; OD, Optical density; RPMI, Roswell Park Memorial Institute medium; FCS, fetal calf serum; MTT, 3-[4,5-dimethylthiazol-2-yl]-2,5-diphenyltetrazolium bromide; HPLC, High Performance Liquid Chromatography; vvm, volume of gas per volume of liquid per minute.

accessible sources of novel bioactive metabolites as we can see with *Streptomyces* and *Nocardia* strains isolated from underground lake and moonmilk [8].

Microbial population associated with lichens have been recently characterized in the lichen symbiosis classically defined like a dual symbiosis between a green alga (or a cyanobacterium) and a fungus [9]. Indeed, a plethora of bacteria have been identified from lichens using culture-independent (reviewed by Suzuki et al [10]) or culture-dependent approaches [11,12] and their bacterial community seems to be specific to the lichen species [13]. This long-lasting ecological niche is a novel reservoir for interesting bacterial strains. The three main bacterial phyla associated with lichens are Proteobacteria, Firmicutes and Actinobacteria [12–14]. The chemical production of several strains associated to lichens has already been described [10,15] highlighting their ability to produce active compounds.

In this study, an Actinobacterium isolated from a terrestrial lichen species *Lathagrium auriforme* [12] (ex *Collema auriforme*), close to *Nocardia* sp. (100% 16S rRNA sequence similarity with *Nocardia soli*, *N. cummidelens* and *N. salmonicida* and 98.59% with *N. ignorata*), was selected. To our knowledge only our previous study has described the chemical production of this *Nocardia* strain [16]. In our attempts to focus on the discovery of novel agents to treat melanoma, one of the most fastest growing forms of cancer, we have selected the culture broth extract of this bacterium which showed an interesting cytotoxic activity against murine B16 melanoma cell lines (IC$_{50}$ = 23 ± 3 μg/mL). In the previous work, the cytotoxic compound(s) involved in the biological activity of this strain were not identified. In this context, the optimization of culture conditions leading to the increase of the cytotoxicity of the bacterial broth was necessary to highlight bioactive compound(s). The aim of this study was to estimate the impact of different culture parameters on this cytotoxicity. We have chosen herein to evaluate various parameters: pH, temperature, stirring speed and the inoculum ratio (%) (v/v). The evaluation of all the possible combinations of these parameters requires 81 experiments, which is very time consuming. A Taguchi L9 orthogonal array design, which is a robust statistical DOE (Design of Experiments) method, allowing to study a set of variables with a limited number of trials, was employed to perform 9 experiments instead of 81 [17]. At the end of the stationary growth phase the cultures were stopped and we determined the yield of extraction and the cytotoxic activity of the extracts against a cancer cell line (B16) and a non-cancer cell line (HaCaT) which will eventually led us to determine a selectivity of action. ANOVA (Analysis Of Variance) test was used to highlight which parameters exhibited a significant influence on the cytotoxicity of the bacterial extracts. Finally, the chemical analyses of the most active extracts were also carried out using HPLC (High Performance Liquid Chromatography) in order to highlight the common metabolites.

## Materials and methods

### Microorganism

The strain used for this study was isolated from a terrestrial lichen *Lathagrium auriforme* (ex *Collema auriforme*) collected in Kesselfallklamm in Austria (47˚12'21.26" N, 15˚23'57.27" E) in November 2012 by Parrot et al [12]. Its 16S rRNA gene was then sequenced using Sanger sequencing (874 pb) and the close phylogenetic neighbors of our strain were identified as *Nocardia ignorata* DQ659907 at 98.59% sequence identity by comparison of these data with sequences in the Eztaxon server type strain database [18]. A second sequencing of 993 pb following the same protocol as described by Parrot et al [12] allowed the identification of three closer phylogenetic neighbors *Nocardia soli*, *N. cummidelens* and *N. salmonicida* at 100% sequence identity using the Eztaxon server. The strain was stored after growth in ISP2 medium [19] with 50% v/v glycerol or 5% v/v DMSO (Dimethylsulfoxyde) at −80˚C and referenced as

DP94 [12]. This strain was deposited at the Institute of Plant Sciences, University of Graz, Austria.

## Preliminary assays

**Precultures.** The precultures were obtained by inoculation of 30 mL of medium with an isolated colony collected on an agar plate. The culture was incubated until the optical density reached 0.6 for further uses. A purity control was performed on agar plate to check the absence of contamination.

**Small-scale cultures.** Four media cited below were used for preliminary assays:

- TY (Tryptone Yeast medium): 10 g/L yeast extract (Sigma-Aldrich, St Louis, MO, USA), 16 g/L tryptone (Sigma-Aldrich, St Louis, MO, USA), and 5 g/L NaCl (Sigma-Aldrich, St Louis, MO, USA),

- Modified LB (Lysogeny Broth) (LBm): 5 g/L peptone (Sigma-Aldrich, St Louis, MO, USA), 5 g/L NaCl (Sigma-Aldrich, St Louis, MO, USA) and 3 g/L yeast extract (Sigma-Aldrich, St Louis, MO, USA)

- ISP2 (International Streptomyces Project medium 2): 4 g/L yeast extract (Sigma-Aldrich, St Louis, MO, USA), 10 g/L malt extract (Sigma-Aldrich, St Louis, MO, USA) and 4 g/L glucose

- MB (Marine Broth): 37.4 g/L of commercial marine broth (Difco ® 2216, ThermoFisher, Waltham, MA, USA)

30 mL of each medium were inoculated in duplicate with the bacterial strain and incubated at 25°C and at 110 rpm in an incubator New Brunswick® Innova 42 (New Brunswick®, Edison, NJ, USA). After 8, 11 or 14 days of incubation the culture broth was centrifuged at 3000 rpm (Thermo scientific Sorvall ST40R, ThermoFisher scientific, Waltham, MA, USA), at 4°C for 15 min. The supernatant was extracted with 2 x 30 mL of ethyl acetate and the organic layer was dried on anhydrous sodium sulfate and the solvent was evaporated *in vacuo* leading to a raw extract.

**Scale-up process.** 5.4 L of TY medium and 7.2 L of LBm medium were divided in 500 mL Erlenmeyer flasks and inoculated with 1% of preculture of the bacterial strain. The cultures were incubated at 25°C, 110 rpm in an incubator New Brunswick® Innova 42 (New Brunswick®, Edison, NJ, USA). After 11 days of incubation the culture broth was extracted as described below. The monitoring of the bacterial growth of these scale-up cultures is reported in S1 Fig.

## Culture extraction

After fermentation, the broth was collected and centrifuged at 3000 rpm (Thermo scientific Sorvall ST40R, ThermoFisher scientific, Waltham, MA, USA), at 4°C for 15 min. The supernatant was collected and 40 g of a resin Amberlite® XAD 7HP (Sigma-Aldrich, St Louis, MO, USA) were added per liter. After 4h of stirring at 150 rpm the resin was filtered and desorbed three times successively with 400 mL of a mixture of MeOH/acetone (1/1, v/v) per 40 g of resin. The solvents were evaporated *in vacuo* and the residue was dissolved in 100 mL of water and extracted with 2 x 100 mL of EtOAc. The organic layer was then dried with anhydrous sodium sulfate, filtered and evaporated *in vacuo* leading to the resin extract (RE). The supernatant previously treated with the resin was extracted two times with EtOAc (1/2, v/v) and the organic layer was dehydrated with anhydrous sodium sulfate, filtered and dried *in vacuo*

leading to the supernatant extract (SE). An extraction yield was defined as follows:

$$\text{Yield} = \frac{\text{masse of the extract (mg)}}{\text{volume of culture medium (L)}}$$

## Culture in bioreactor

Culture was performed in 2.5 L of TY medium in a bioreactor BioFlo/Celligen® 115 (New Brunswick®, Edison, NJ, USA). The oxygenation rate of the medium was regulated at 0.4 vvm (volume of gas per volume of liquid per minute) and the percentage of dissolved oxygen was maintained up to 40% by using the cascade mode with the stirring. Temperature, pH, stirring and inoculum ratio (%) (v/v) were selected according to the design of experiments (Tables 1 and 2). The cultures were stopped after 5 days of fermentation. An example of monitoring of culture parameters and of bacterial growth has been reported in S2 and S3 Figs.

## Culture in Erlenmeyer flasks

Culture of the strain was performed in 300 mL of TY medium in 500 mL Erlenmeyer flask. The pH of the medium was adjusted with a solution of NaOH 5 M or $H_2SO_4$ 0.5 N. Flasks were then incubated in an incubator New Brunswick® Innova 42 (New Brunswick®, Edison, NJ, USA) with a stirring fixed at 150 rpm and a temperature fixed at a value according to the design of experiments (See Tables 3 and 4). The cultures were stopped after 11 days of fermentation. The monitoring of bacterial growth was performed by measurement of OD (Optical Density) as shown in S4 Fig.

## Taguchi experimental design

The Taguchi method was selected to establish the experimental design and led to 9 experiments either in bioreactor (Tables 1 and 2) nor in Erlenmeyer flask (Tables 3 and 4). For each experiment, the responses evaluated were the yield and the cytotoxicity of extracts against B16 and HaCaT cell lines. For the fermentation in bioreactor we selected 4 independent parameters and attributed 3 levels for each. The four parameters selected were temperature (25°C, 30°C and 37°C), pH (6, 7 and 8), percentage of inoculum (v/v) (1%, 2% and 5%) and stirring speed (150, 200 and 250 rpm) (Table 1). The experimental design and ANOVA analysis were performed using Excel 2016.

For the fermentation in Erlenmeyer flasks the stirring was fixed at 150 rpm and only 3 parameters were selected with 3 levels for each (Tables 3 and 4).

## Cytotoxic assays

The cytotoxic activity of the resin extracts and the supernatant extracts was evaluated for each condition of the experimental design with a standard tetrazolium assay (MTT) [20]. Two cell lines (B16: murine melanoma, HaCaT: human keratinocytes) were cultivated in RPMI (Roswell Park Memorial Institute) 1640 medium (Thermo Fisher Scientific, USA) supplemented

**Table 1. Assignment of factors and level setting of the orthogonal array design L9 ($3^3$) for cultures in bioreactor.**

| Factors | Level 1 | Level 2 | Level 3 |
|---|---|---|---|
| [A] Temperature (°C) | 25 | 30 | 37 |
| [B] pH | 6 | 7 | 8 |
| [C] inoculum ratio % | 1 | 2 | 5 |
| [D] stirring speed (rpm) | 150 | 200 | 250 |

**Table 2. Assignment of the experimental conditions in the orthogonal design L9 ($3^4$) for cultures in bioreactor.**

| Exp n° | Factor[a] | | | |
|---|---|---|---|---|
| | **[A]** | **[B]** | **[C]** | **[D]** |
| **B1** | 1 | 1 | 1 | 1 |
| **B2** | 2 | 1 | 2 | 2 |
| **B3** | 3 | 1 | 3 | 3 |
| **B4** | 1 | 2 | 2 | 3 |
| **B5** | 2 | 2 | 3 | 1 |
| **B6** | 3 | 2 | 1 | 2 |
| **B7** | 1 | 3 | 3 | 2 |
| **B8** | 2 | 3 | 1 | 3 |
| **B9** | 3 | 3 | 2 | 1 |

[a][A]: Temperature, [B]: pH, [C]: inoculum ratio (%), [D]: stirring

with 5% fetal calf serum (FCS) and 1% antibiotics (penicillin 10 000 UI/ml and streptomycin 10 000 μg/ml, Eurobio®, Les Ulis, France). Cells were then seeds in 96-wells plate (B16: 6000 cells/well and HaCaT 10,000 cells/well) at day 0 and incubated for 24h at 37°C and 5% $CO_2$. Extracts were then added at different concentrations (1, 10, 50, 100, and 200 μg/mL) for all the experiments. For experimental design in bioreactor and Erlenmeyer flasks the extracts were tested a second time at 5, 50, 100, 250 and 500 μg/mL to expand the range of $IC_{50}$ values. After 24h of incubation (37°C, 5% $CO_2$) the cell viability was measured at 540 nm using a MTT (3-[4,5-dimethylthiazol-2-yl]-2,5-diphenyltetrazolium bromide) assay [20]. Each experiment was repeated three times.

## Chemical analyses

Chemical analyses of the most active extracts were carried out on a HPLC system—Diode Array Detector (LC-DAD) (Shimadzu, Marne La Vallée, France). A Prevail C18 column (5 μm, 250 × 4.6 mm, GRACE, Columbia, MD, USA) was used and a gradient system was applied: A (0.1% formic acid in water) and B (0.1% formic acid in acetonitrile). The following gradient was applied at a flow rate of 0.8 mL/min in the HPLC system: initial: 100% (A); from 0 to 5 min: 100% (A); from 5 to 35 min: 100% (A)/0% (B) to 0% (A)/100% (B); from 35 to 45 min: 100% B; from 45 to 50 min: 0% (A)/100% (B) to 100% (A)/0% (B); from 50 to 55 min: 100% (A). Samples were prepared by dissolving extracts in MeOH at 1 mg/mL and 20 μL were injected after filtration (45 μm). The LabSolutions software (Shimadzu, Marne La Vallée, France) was used for data analyses.

## Results

### Preliminary assays

The *Nocardia* sp. strain used in this study achieved the stationary growth phase after 7 days of growth. We have cultivated this strain using 30 mL of four different media in 50 mL tubes,

**Table 3. Assignment of factors and level setting of the orthogonal array design L9 ($3^3$) for cultures in Erlenmeyer flasks.**

| Factors | Level 1 | Level 2 | Level 3 |
|---|---|---|---|
| **[A] Temperature (°C)** | 25 | 30 | 37 |
| **[B] pH** | 6 | 7 | 8 |
| **[C] inoculum ratio %** | 1 | 2 | 5 |

**Table 4. Experimental conditions in the orthogonal design L9 ($3^3$) for cultures in Erlenmeyer flasks.**

| Exp n° | Factor[a] | | |
|---|---|---|---|
| | [A] | [B] | [C] |
| E1 | 1 | 1 | 1 |
| E2 | 1 | 2 | 2 |
| E3 | 1 | 3 | 3 |
| E4 | 2 | 1 | 2 |
| E5 | 2 | 2 | 3 |
| E6 | 2 | 3 | 1 |
| E7 | 3 | 1 | 3 |
| E8 | 3 | 2 | 1 |
| E9 | 3 | 3 | 2 |

[a][A]: Temperature, [B]: pH, [C]: inoculum ratio (%)

with a 7 days of exponential growth phase followed by 1, 4 or 7 days of stationary phase (respectively 8, 11 and 14 days in total). The best cytotoxic activity against B16 cells was achieved with the culture in LBm medium for 11 days ($IC_{50}$ = 8 ± 2 μg/mL) and with the ISP2 medium after 14 days of culture ($IC_{50}$ = 7.5 ± 0.5 μg/mL). Similar activities have been shown on HaCaT cells mentioning no selectivity of action. The highest amount of extract was obtained when the strain was grown in TY medium (Table 5). A scale-up assay with 300 mL of media in 500 mL Erlenmeyer flasks was then performed using LBm and TY for 11 days of culture (4 days of stationary phase). In order to have sufficient amount of LBm extracts and based on the results obtained with the 30 mL process, we have inoculated 24 Erlenmeyer flasks with LBm medium (total volume 7.2 L) and 18 Erlenmeyer flasks with TY medium (total volume 5.4 L). The culture extract from TY medium gave the best activity against B16 cell line ($IC_{50}$ = 4.6 ± 1.8 μg/mL) even though the amount obtained was low (yield = 16.1 mg/L) but higher than for the culture in LBm (Table 6). These culture conditions (TY medium and 4 days of stationary phase) were applied to the Taguchi design experiments.

## Experimental design in bioreactor

An experimental design was applied in order to find the best culture conditions to improve the yield and the cytotoxic activity of extracts. Varying temperature [A] (25, 30 and 37°C), pH [B] (6, 7 and 8), percentage of inoculum ratio (% v/v) [C] (1, 2 and 5%) and stirring speed [D] (150, 200 and 250 rpm), 9 experiments were carried out. In all experiments, the yield of the resin extracts (RE) and the supernatant extracts (SE) were determined and their cytotoxicity was measured against B16 and HaCaT cell lines (Table 7). When similar activities have been exhibited against the two cell lines, we have focused on activities against B16.

The mean $IC_{50}$ values of RE on B16 cells, were classified into three groups: a first group with $IC_{50}$ > 300 μg/mL (exp. **B5** and **B8**); a second group with 100 < $IC_{50}$ < 300 μg/mL (exp. **B1**, **B2** and **B7**) and the last group with $IC_{50}$ < 100 μg/mL (exp. **B3**, **B4**, **B6** and **B9**) (Table 7). In a similar manner, the results were classified into three groups for SE extracts based on their activity on B16: the first group with $IC_{50}$ > 400 μg/mL (exp. **B1**, **B2** and **B7**); a second group with 100 < $IC_{50}$ < 400 μg/mL (exp. **B4**, **B6** and **B9**) and the last group with $IC_{50}$ < 100 μg/mL (exp. **B3**, **B5** and **B8**) (Table 7).

Most of RE extracts exhibited cytotoxic activities whereas only a few of SE extracts were active. Main-effect plots (Fig 1) indicated that the $IC_{50}$ values decreased with the maximum

**Table 5. Weight and cytotoxicity of extracts in preliminary assays.**

| Days of culture | Medium[a] | Raw extract amount (mg) | IC$_{50}$ (µg/mL) | |
|---|---|---|---|---|
| | | | B16 | HaCaT |
| 8 | ISP2 | 2.3 ± 0.0 | 45 ± 5 | 59 ± 5 |
| | TY | 5.2 ± 0.1 | 41 ± 5 | 72 ± 7 |
| | MB | 2.2 ± 0.3 | 105 ± 20 | 115 ± 15 |
| | LBm | 2.5 ± 0.4 | 25 ± 4 | 95 ± 30 |
| 11 | ISP2 | 4.0 ± 0.7 | 31 ± 10 | 64 ± 28 |
| | **TY** | **5.4 ± 0.9** | **25 ± 4** | **47 ± 13** |
| | MB | 2.1 ± 0.7 | 57 ± 18 | 120 ± 15 |
| | **LBm** | **2.5 ± 0.1** | **8 ± 2** | **31 ± 12** |
| 14 | **ISP2** | **3.6 ± 0.2** | **7.5 ± 0.5** | **41 ± 5** |
| | TY | 6.4 ± 0.3 | 35 ± 8 | 86 ± 42 |
| | MB | 3.8 ± 0.2 | 85 ± 10 | 79 ± 9 |
| | LBm | 4.2 ± 0.2 | 14 ± 6 | 70 ± 4 |

[a]ISP2 = International Streptomyces Project 2, TY = Tryptone Yeast, MB = Marine Broth, LBm = modified Lysogeny Broth

level of factor B at 37°C. It was also noticed that the IC$_{50}$ values increased in culture extracts at 30°C, highlighting the importance of culture temperature on the cytotoxicity of the extracts.

Finally, the experiment with the highest cytotoxic activity was the experiment **B3** (RE, IC$_{50}$ = 53 ± 16 µg/mL and SE IC$_{50}$ = 41 ± 8 µg/mL on B16) with the following culture conditions: pH 6, temperature 37°C, an inoculum of 5% and a stirring of 250 rpm.

Surprisingly, the experiment **B2** (culture conditions: pH 6, temperature of 30°C, inoculum of 2% and stirring speed of 200 rpm) gave the highest supernatant extraction (SE) yield.

Based on the cytotoxic activity against B16 and the extraction yield, ANOVA test was used to highlight which parameter(s) could significantly influence the production of active compound(s).

**Statistical analysis.** The F-ratio and the p-value determined from the ANOVA analysis on the cytotoxicity of the resin extracts showed that the temperature significantly influenced the activity ($p < 0.05$) (Table 8 and Fig 1). Conversely, none of the parameters selected for this analysis showed a significant effect on the extraction yield as well as for the cytotoxicity of the supernatant extracts (Table 9).

**Validation of the optimal conditions.** The experiment **B3** with the following parameters, pH 6, temperature 37°C, inoculum 5% and stirring 250 rpm, was repeated once using bioreactor to confirm the previous results on HaCaT and B16. The repetition of this experiment showed close values of IC$_{50}$ for SE on B16 and HaCaT cell lines (IC$_{50}$ = 33 ± 8 $\mu$g/mL on B16 and IC$_{50}$ = 23 ± 5 $\mu$g/mL on HaCaT) and a decrease of the activity of RE against B16 (IC$_{50}$ > 200 $\mu$g/mL). This experiment confirms nevertheless that the culture conditions used are optimal for the cytotoxicity of the extracts from bioreactor.

**Table 6. Extraction yield and cytotoxicity of extracts from scale up in Erlenmeyer flasks.**

| Medium | Extract | Yield (mg/L) | IC$_{50}$ (µg/mL) | |
|---|---|---|---|---|
| | | | B16 | HaCaT |
| TY | Resin extract | 101.4 | >200 | >200 |
| | Supernatant extract | 16.1 | **4.6 ± 1.8** | **8 ± 1.5** |
| LBm | Resin extract | 60.3 | >200 | >200 |
| | Supernatant extract | 3.0 | 52 ± 20 | 73 ± 20 |

**Table 7. Extraction yield and cytotoxicity of the extracts from the experimental design using bioreactor.**

| Exp n˚ | RE | | | SE | | |
|---|---|---|---|---|---|---|
| | Yield (mg/L) | IC$_{50}$ (µg/mL) | | Yield (mg/L) | IC$_{50}$ (µg/mL) | |
| | | B16 | HaCaT | | B16 | HaCaT |
| B1 | 146.1 | 157 ± 77 | 92 ± 29 | 46.3 | >500 | 460 ± 140 |
| B2 | 160.5 | 205 ± 100 | 185 ± 65 | 1859.3 | >500 | >500 |
| B3 | **216.6** | **53 ± 16** | **152 ± 28** | 478.6 | **41 ± 8** | **20 ± 8** |
| B4 | 158.0 | 92 ± 19 | 70 ± 8 | 152.3 | 250 | 320 |
| B5 | 177.8 | 345 ± 95 | 120 ± 40 | 506.9 | 100 ± 30 | 150 |
| B6 | 223.5 | 45 ± 15 | 80 ± 20 | 36.1 | 270 ± 35 | 225 ± 45 |
| B7 | 126.8 | 270 ± 30 | 185 ± 15 | 66.5 | 430 ± 90 | 180 ± 20 |
| B8 | 703.0 | 340 ± 25 | 397 ± 27 | 18.4 | 67 ± 4 | 38 ± 19 |
| B9 | 318.0 | 90 ± 50 | 110 ± 25 | 43.9 | 350 | 270 |

## Experimental design in Erlenmeyer flasks

The data obtained in Erlenmeyer flasks were analyzed in a same manner using a Taguchi array and the yields and cytotoxic activities of the both extracts (RE and SE) were measured on B16 and HaCaT cell lines (Table 10).

The most important cytotoxic activity was demonstrated with experiment **E7** (RE, IC$_{50}$ = 6 ± 0.5 µg/mL on B16) with the following culture conditions: pH 6, temperature 37˚C and 5% inoculum and led to a sufficient amount of extract (202.0 mg/L) (Table 10). Interestingly, these conditions were close to those obtained for the culture in bioreactor (excepted for the value of stirring fixed at 150 rpm for Erlenmeyer process), confirming that these parameters are the best to improve the cytotoxicity of the extracts. Conversely, the highest extraction yield (840.3 mg/mL) was obtained with **E8** and the conditions (37˚C, pH 7 and inoculum 1%, Table 10) were completely different than those obtained for culture in bioreactor.

Nevertheless, the ANOVA analysis of the results obtained from these Erlenmeyer cultures showed that none of the selected parameters had a significant effect on the cytotoxicity of the extracts on B16 cell lines or on the extraction yield.

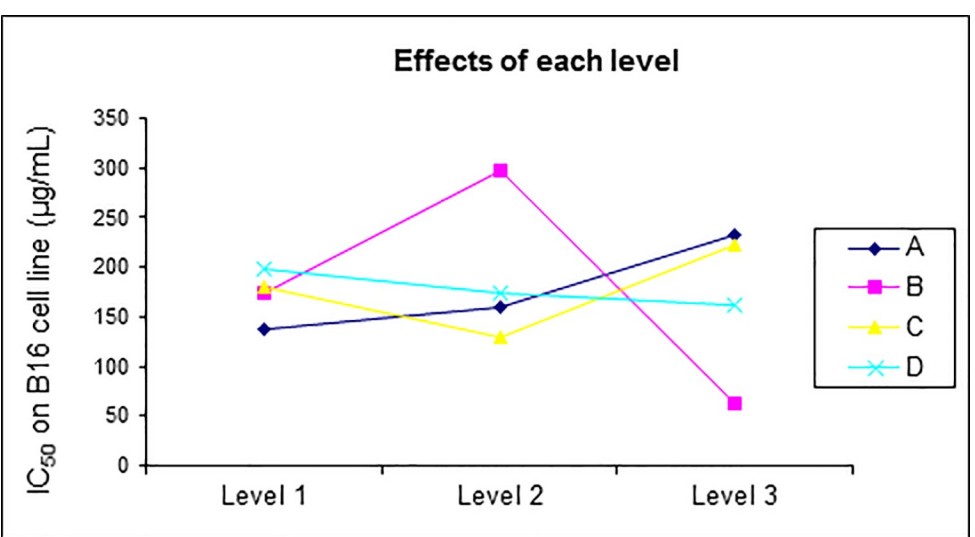

**Fig 1. Main-effect plots for optimization of culture conditions in bioreactor.**

**Table 8. Summary of analysis of variance in the ANOVA test for cytotoxicity on B16 of RE from bioreactor.**

| Factors[a] | Sum of squares | f | Average of square | F-ratio | F0.95(2,2) | p |
|---|---|---|---|---|---|---|
| A | 14804.2 | 2 | 7402.1 | 7.46 | 19 | 0.118 |
| B | 82222.9 | 2 | 41111.4 | 41.44 | 19 | 0.024 |
| C | 1984.2 | 2 | 992.1 | 1.00 | 19 | 0.500 |
| D | 13206.9 | 2 | 6603.4 | 6.66 | 19 | 0.131 |
| Error | 1984.2 | 2 | 992.1 | | | |
| Total | 112218.2 | | | | | |

[a] [A]: Temperature, [B]: pH, [C]: inoculum ratio (%), [D]: stirring

## Impact of the culture vessel on the activity of extracts

In order to determine the effect of the culture conditions on the production of cytotoxic metabolites, we compared the results obtained for three processes performed using either bioreactor or Erlenmeyer flasks and possessing similar culture conditions with a stirring speed fixed at 150 rpm (Table 11). Among them, the most active extracts were obtained at 37°C, pH 6 and a 2% inoculum ratio (experiments **B9** and **E9**). The yields were also higher in these experiments in comparison to the other ones.

## Comparison of HPLC profiles of the most active extracts

The HPLC chromatograms of these two experiments **B9** and **E9** were compared in Fig 2. Those of the most active extracts obtained during this study (SE of the large scale culture in TY shown in preliminary assays, SE from **B3** and RE from **E7**) were reported in Fig 3. The compounds previously isolated from this *Nocardia* strain [16] and known compounds such as cyclo (*L*-Ala-*L*-Phe) [21], adenine, adenosine and 1-(5-deoxy-β-D-erythro-pent-4-enofurano-syl) [22] were identified by comparison with a standard and data from literature (Figs 2A, 2B and 3).

The main differences between the production of metabolites by the strain in the RE of **B9** and **E9** are due to the presence of an unidentified compound with a r.t. of 19 min and of a diketopiperazine cyclo-(*L*-Ala-*L*-Phe) which were produced in large amount in bioreactor **B9** but not in Erlenmeyer flask **E9** (Fig 2A). Variability was also observed for the production of compounds eluted between 26 and 32 min (Fig 2A). In the SE of **E9** a compound eluted at the beginning of the gradient appeared. Moreover, a difference of the production of compounds eluted between 22 and 28 min was also highlighted between the two culture vessels (Fig 2B). The production of adenine was most important in bioreactor than in Erlenmeyer reaction set-up (Fig 2). Despite these differences in the chemical profiles, the activity of the extracts were

**Table 9. Summary of p-value from the ANOVA test on all extracts from bioreactor.**

| Factors[a] | p | | | |
|---|---|---|---|---|
| | RE | | SE | |
| | Effect on IC$_{50}$ on B16 | Effect on extraction yield | Effect on IC$_{50}$ on B16 | Effect on extraction yield |
| A | 0.118 | 0.408 | 0.500 | 0.303 |
| B | **0.024** | 0.475 | 0.333 | 0.312 |
| C | 0.500 | 0.490 | 0.192 | 0.500 |
| D | 0.131 | 0.500 | 0.388 | 0.385 |

[a] [A]: Temperature, [B]: pH, [C]: inoculum ratio (%), [D]: stirring

**Table 10. Extraction yield and cytotoxicity of the extracts on B16 and HaCaT cell lines for the experimental design in Erlenmeyer flasks.**

| Exp n° | RE | | | SE | | |
|---|---|---|---|---|---|---|
| | Yield (mg/L) | IC$_{50}$ (µg/mL) | | Yield (mg/L) | IC$_{50}$ (µg/mL) | |
| | | B16 | HaCaT | | B16 | HaCaT |
| E1 | 306.7 | 280 ± 160 | 110 ± 60 | 199.3 | 375 ± 110 | 40 ± 15 |
| E2 | 279.7 | 96 ± 9 | 122 ± 4 | 77.9 | 73 ± 20 | 77 ± 16 |
| E3 | 218.7 | 70 ± 8 | 71 ± 19 | 66.0 | 74 ± 34 | 56 ± 24 |
| E4 | 96.3 | 145 ± 15 | 122 ± 22 | 74.0 | 380 ± 105 | 75 ± 50 |
| E5 | 184.0 | 130 ± 4 | 127 ± 13 | 57.3 | 390 ± 100 | 240 ± 150 |
| E6 | 305.7 | 64 ± 8 | 72 ± 1 | 58.3 | 170 ± 10 | 152 ± 18 |
| E7 | 202.0 | **6 ± 0.5** | 30 ± 7 | 583.3 | 480 ± 230 | 140 ± 70 |
| E8 | 176.7 | 81 ± 10 | 50 ± 7 | 840.3 | >500 | 260 ± 110 |
| E9 | 489.0 | 63 ± 7 | 79 ± 15 | 39.7 | 137 ± 7 | 32 ± 10 |

similar between bioreactor and Erlenmeyer flasks (except for the SE on B16 cells) meaning that the compounds highlighted above are not implicated in the cytotoxicity of the extracts.

Finally, the comparison of the most active extracts led to the highlighting of unidentified compounds common in all the three active extracts which were indicated with black squares in Fig 3.

## Discussion

Lichens are a novel ecological niche of considerable interest for the discovery of bacterial strains with biotechnological potential [12]. We selected and studied a lichen-associated Actinobacterium, *Nocardia* sp that demonstrated interesting cytotoxic activities. The results detailed herein showed that the production of cytotoxic compounds by this *Nocardia* sp. could be modulated by culture conditions according to the OSMAC (One Strain/Many Compounds) approach previously described [23]. The data obtained in preliminary assays are in accordance with our expectations. The marine broth currently used in our experiments performed on strains isolated from marine lichens [12,24] is not suitable to optimize the cytotoxicity of our *Nocardia* strain isolated from a terrestrial lichen. Moreover, the amount of production of metabolites increases with rich medium such as TY. This observation is close to those of Abdelmohsen et al [25] who indicated that the production of actinosporin by a marine sponge associated-Actinokineospora was widely affected depending on the culture medium used. Our results suggest in the same manner that the production of the compound(s) involved in the cytotoxicity of the extracts is influenced by the difference in the medium growth used.

**Table 11. Comparison of the results obtained for cultures in bioreactor and in Erlenmeyer flasks with the same culture conditions.**

| Experiences | Factors | | | RE | | | SE | | |
|---|---|---|---|---|---|---|---|---|---|
| | Temperature (°C) | pH | Inoculum ratio (%) | Yield (mg/L) | IC$_{50}$ (µg/mL) | | Yield (mg/L) | IC$_{50}$ (µg/mL) | |
| | | | | | B16 | HaCaT | | B16 | HaCaT |
| B1 | 25 | 6 | 1 | 146.1 | 157 ± 77 | 92 ± 29 | 46.3 | >500 | 460 ± 140 |
| E1 | 25 | 6 | 1 | 306.7 | 280 ± 160 | 110 ± 60 | 199.3 | 375 ± 110 | 40 ± 15 |
| B5 | 30 | 7 | 5 | 177.8 | 345 ± 95 | 120 ± 40 | 506.9 | 100 ± 30 | 39 ± 10 |
| E5 | 30 | 7 | 5 | 184.0 | 130 ± 4 | 127 ± 13 | 57.3 | 390 ± 100 | 240 ± 150 |
| B9 | 37 | 8 | 2 | **318.0** | **90 ± 50** | **110 ± 25** | **43.9** | **390 ± 90** | **45 ± 2** |
| E9 | 37 | 8 | 2 | **489.0** | **63 ± 7** | **79 ± 15** | **39.7** | **137 ± 7** | **32 ± 10** |

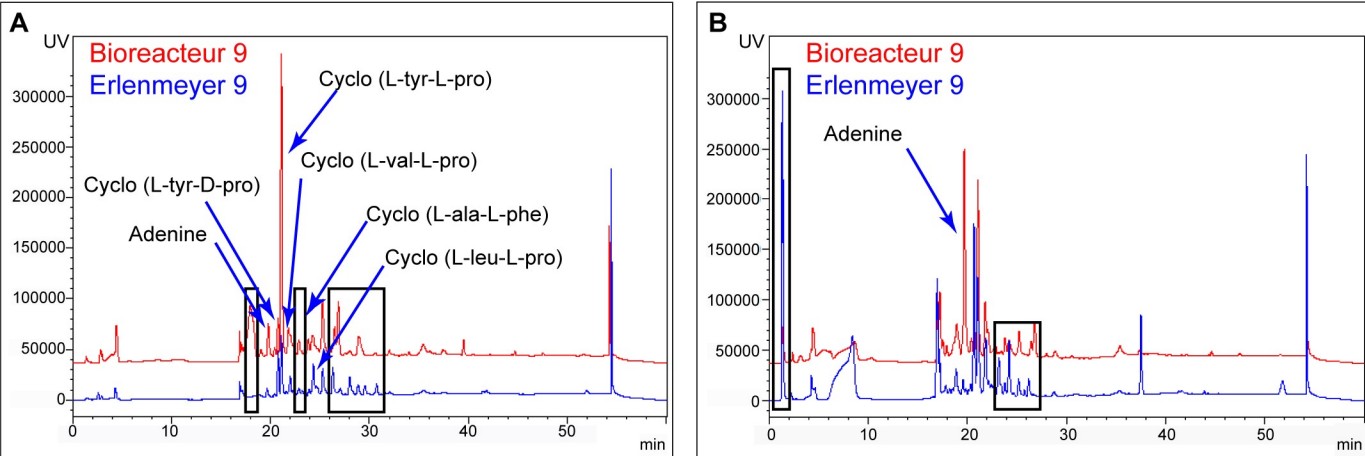

**Fig 2. Comparison of chemical profiles of B9 (red) and E9 (blue).** A. Resin extracts, B. Supernatant extracts. All samples were analysed at 220 nm on Prevail® reversed phase C18 column with a gradient of $H_2O$ (A)/acetonitrile (B) (10 min 100% of A, 30 min from 0% of B to 100% of B, 10 min 100% of B).

A significant effect of temperature on cytotoxicity of the resin extracts ($p < 0.05$) with an improvement of the activity at 37˚C was exhibited by ANOVA analysis of the data from experimental design in bioreactor. The importance of culture parameters such as temperature on increasing of the biomass and of the activity has been already demonstrated for Actinobacteria

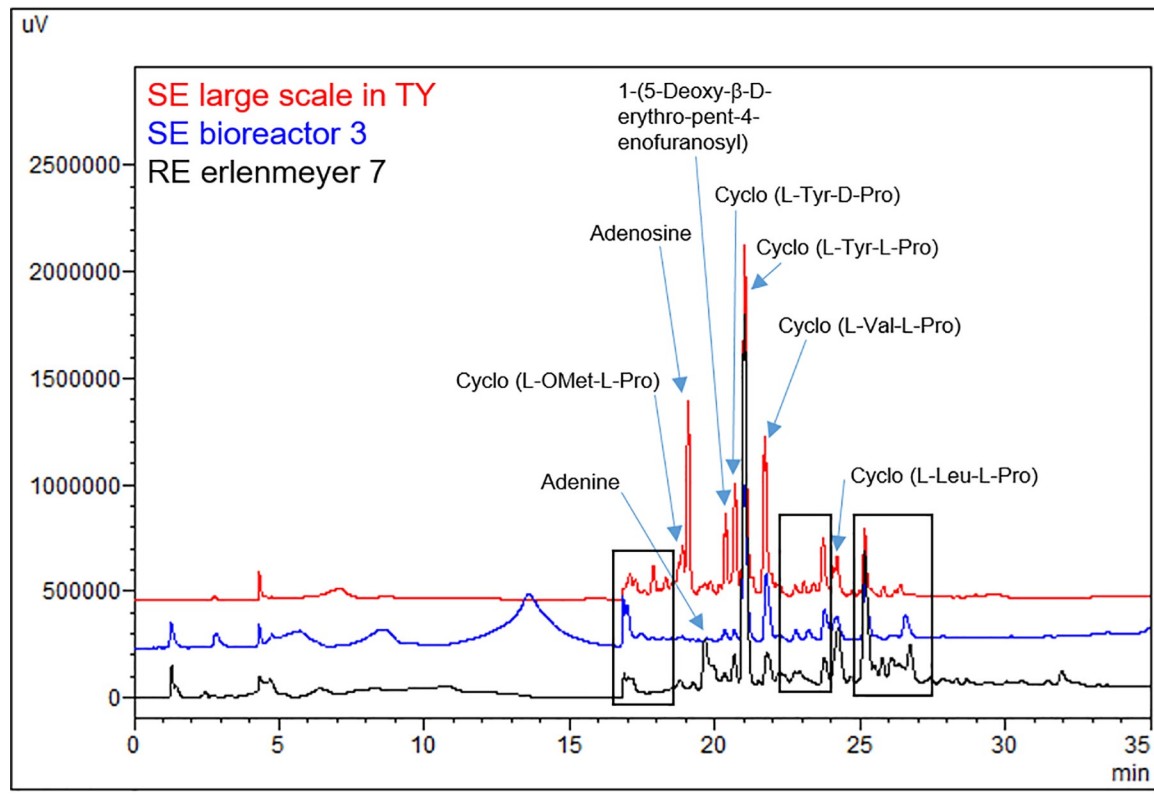

**Fig 3. Comparison of the chemical profiles of the most active extracts.** SE of the large scale culture in TY shown in preliminary assays (red), SE of **B3** (blue) and RE of **E7** (black). All samples were analysed at 220 nm on Prevail® reversed phase C18 column with a gradient of $H_2O$ (A)/acetonitrile (B) (10 min 100% of A, 30 min from 0% of B to 100% of B, 10 min 100% of B).

strains. Thus, the effect of the temperature has been described on the biomass and geosmin production by strains of *N. cummidelens* and *N. fluminea*. The highest biomass production was thus obtained at the warmest temperature tested, 30˚C and 25˚C, for the both strains and the production of geosmin was optimal at 25˚C [26]. Similar effects has been reported on the production of rapamycin by a *Streptomyces* strain, where the optimal temperature was established at 23˚C, the lowest tested value [27].

In spite of employing similar culture conditions (temperature 37˚C, pH 6 and inoculum 5%,) in both the bioreactor (**B3**) and Erlenmeyer flasks (**E7**), the best activity (IC$_{50}$ = 6 ± 0.5 μg/mL on B16) was displayed by extracts obtained from Erlenmeyer flasks (**E7**) highlighting the importance of the culture process used.

Various compounds have been previously isolated from the culture broth of this bacterial strain [16]. Compounds identified were mainly diketopiperazines. An auxin derivative and purine derivatives (e.g. adenine) have also been isolated (data not published) but none of these compounds explained the cytotoxic activity of the bacterial extracts.

The comparison of the chemical profiles of extracts obtained by culture in bioreactor and in Erlenmeyer flasks highlighted several differences which also revealed the impact of the system used. These variations of production could be explained by the different aeration and the stirring system used [28] leading to different stress conditions for bacteria.

The comparison of the chemical profiles of the most active extracts (Fig 3) showed a very similar production and the major compounds already isolated possess no cytotoxic properties on B16 cell line. Nevertheless, some zones of the chromatogram corresponding to unidentified compounds have been highlighted (Fig 3).

## Conclusion

In conclusion, the Taguchi method for design of experiments combined with an ANOVA test were shown to be powerful tools for the optimization of culture parameters of lichen-associated *Nocardia* sp. for the production of cytotoxic compounds. Further separative experiments will be carried out with a focus on the zones highlighted on HPLC profiles during this study in order to identify compound(s) involved in the cytotoxic activity of these bacterial extracts.

## Supporting information

**S1 Fig. Monitoring of bacterial growth of scale-up in TY and LBm.**
(TIF)

**S2 Fig. Monitoring of culture parameters of B1.**
(TIF)

**S3 Fig. Monitoring of bacterial growth of B1.**
(TIF)

**S4 Fig. Monitoring of bacterial growth of E1.**
(TIF)

## Acknowledgments

We sincerely thank L. Intertaglia (LBBM, Banyuls/mer) for the identification of the strain DP94 and Dr S. Sabbani for the reading and the improvement of English langage.

## Author Contributions

**Conceptualization:** Alba Noël, Sophie Tomasi.

**Formal analysis:** Eric Hitti.

**Investigation:** Alba Noël, Gwendoline Van Soen, Isabelle Rouaud.

**Methodology:** Isabelle Rouaud.

**Supervision:** Sophie Tomasi.

**Validation:** Eric Hitti, Sophie Tomasi.

**Writing – original draft:** Alba Noël, Sophie Tomasi.

**Writing – review & editing:** Sophie Tomasi.

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
