## [Decision Letter · Decision Letter 0]

16 Sep 2019

PONE-D-19-18890

Optimization of cytotoxic activity of Nocardia sp culture broths using a design of experiments

PLOS ONE

Dear Prof. Tomasi,

Thank you for submitting your manuscript to PLOS ONE. After careful consideration, we feel that it has merit but does not fully meet PLOS ONE’s publication criteria as it currently stands. Therefore, we invite you to submit a revised version of the manuscript that addresses the points raised during the review process.

We would appreciate receiving your revised manuscript by Oct 31 2019 11:59PM. To enhance the reproducibility of your results, we recommend that if applicable you deposit your laboratory protocols in protocols.io, where a protocol can be assigned its own identifier (DOI) such that it can be cited independently in the future. For instructions see: http://journals.plos.org/plosone/s/submission-guidelines#loc-laboratory-protocols

We look forward to receiving your revised manuscript.

Kind regards,

Yoshihiro Uesawa

Academic Editor

PLOS ONE

Journal Requirements:

"We thank the “Ligue contre le cancer” foundation for the financial support which allowed the acquisition of a bioreactor BioFlo 115® in the laboratory. We sincerely thank L.Intertaglia (LBBM, Banyuls/mer) for the identification of the strain DP94 and Dr S. Sabbani for the reading and the improvement of English langage.

" The funders had no role in study design, data collection and analysis, decision to publish, or preparation of the manuscript."

After careful reading of this manuscript and considering the reviewers comments, I suggest authors to consider the reviewer comments and submit the revised version of this manuscript. 

Reviewers' comments:

Reviewer's Responses to Questions

**Comments to the Author**

1. Is the manuscript technically sound, and do the data support the conclusions?

Reviewer #1: Partly

Reviewer #2: Yes

2. Has the statistical analysis been performed appropriately and rigorously? 

Reviewer #1: Yes

Reviewer #2: Yes

3. Have the authors made all data underlying the findings in their manuscript fully available?

Reviewer #1: Yes

Reviewer #2: Yes

4. Is the manuscript presented in an intelligible fashion and written in standard English?

Reviewer #1: No

Reviewer #2: Yes

5. Review Comments to the Author

Reviewer #1: The manuscript presented by Tomasi et al entitled "Optimization of cytotoxic activity of Nocardia sp culture broths using a design of experiments" addresses the impact of culture media, parameters (bioreactor/shake flask) and process parameters (Temperature, pH, inoculation and stirring) on the yield and cytotoxic effects of the culture fluid on two mammalian cell lines. However, the study has been done in details but there are some concerns in the manuscript. The rational of this study and each experiment should be written clearly.

1. Order of authors is different in the beginning table of submitted file compared to first page of manuscript.

2. Abstract:

- The abstract should be revised and included method section. Moreover, it should be well reflected whole significant information of the manuscript. In the abstract, the advantages of this work should be reflected shortly.

3. Introduction:

- In the introduction section the authors should be briefly clarify about the application of recent work and drawbacks.

- (Lines 65 to 68): What is the reason for choosing B16 cell line to check the cytotoxicity? Authors should be explain more about the application such cytotoxic effect of broth extract on clarified targets such as therapeutic purposes.

- In addition, there is no explanation about the HaCaT cell line in introduction section of the manuscript.

- (Lines 69 to 70): Four parameters were targeted to evaluation in this study such as pH, Temperature, stirring speed and inoculum ratio, so please explain that based on which experimental design basically you reached to 81 experiments then in continue you reduced the number to 9 experiment?

4. Materials and method:

- (Line 110): How you ensure that the number of inoculated bacteria (precultured) were same in all experimental conditions? Please provide method for inoculum preparation and the number of bacteria (CFU/ml) added in both small scale and scaled up experimental condition.

- (Line 147): To keep the oxygen on 40% in the bioreactor, the cascade mode with the stirring was selected. Otherwise, based on experimental design the rpm is one of the variables which was adjusted on 150, 200 and 250 in each experimental condition. So the question arises is that when the amount of oxygen is reduced, the rpm may increase to compensate the aeration/oxygenation. Please explain how you justify such rpm variations during the experimental design.

- Please add the software name and version used for experimental design and statistical analysis.

5. Results:

- The quality of figure no. 2 is not good. I suggest having alternate presentation of this figure.

- The number of repetition in validation of the optimal condition and the results of them should be reported clearly in results section of the manuscript.

- The value of “yield” as a parameter should be followed by standard deviation.

6. Discussion:

- According below statement of the manuscript, the differences between bioreactor and shake flask are well known and they are not much comparable to each other. I strongly recommend to revise the discussion part of manuscript.

“In fact, the stirring in bioreactor is performed by agitation blade submerged in the culture broth which implicates hyphal breaks, while in shaken flasks the agitation is performed with an orbital shaker allowing the hyphal formation. In a similar way, the aeration system in bioreactor consisted of a tube submerged in the culture broth that conducted oxygen into the medium causing the formation of bubbles, while in the shaken flasks there is no specific system for aeration. These main differences could explain the variation of bacterial production profile due to a different oxygenation and stirring methods inducing different stress conditions for bacteria.”

7. Minor comments:

- Some minor comment and suggestions are included in the pdf file of manuscript.

Reviewer #2: The work is interesting since explore the culture conditions to have the highest cytotoxic activity from Nocardia sp in bioreactor and flask. Cultures in flask result better since possibly cells are not as affected by hydrodynamic stresses as in bioreactor. The main different peaks separated by HPLC are not the responsible for the cytotoxic activity, so, there is pending to identify the molecules responsible. There have been identified some molecules with cytotoxic activity from Nocardia, so authors can trying to identify if activity could correspond to this kind of molecules.

6. PLOS authors have the option to publish the peer review history of their article (what does this mean?). If published, this will include your full peer review and any attached files.

Reviewer #1: No

Reviewer #2: No

---

## [Author Response · Author response to Decision Letter 0]

5 Nov 2019

RESPONSES TO REVIEWER’S COMMENTS

Rewiever’s 1 comments : 

The manuscript presented by Tomasi et al entitled "Optimization of cytotoxic activity of Nocardia sp culture broths using a design of experiments" addresses the impact of culture media, parameters (bioreactor/shake flask) and process parameters (Temperature, pH, inoculation and stirring) on the yield and cytotoxic effects of the culture fluid on two mammalian cell lines. However, the study has been done in details but there are some concerns in the manuscript. The rational of this study and each experiment should be written clearly.

1. Order of authors is different in the beginning table of submitted file compared to first page of manuscript.

Due to a problem appeared during the submission process an error occured in the order of authors. The good order is: Alba Noël, Gwendoline Van Soen, Isabelle Rouaud, Eric Hitti, Sophie Tomasi.

2.Abstract:

- The abstract should be revised and included method section. Moreover, it should be well reflected whole significant information of the manuscript. In the abstract, the advantages of this work should be reflected shortly.

The abstract has been rewritten p 2 line 18- 42 following the reviewer’s comments.

3. Introduction:

- In the introduction section the authors should be briefly clarify about the application of recent work and drawbacks.

According to the reviewer’s comments it has been clarified p.4 l.89-92 as: “In the previous work, the cytotoxic compound(s) involved in the biological activity of this strain were not identified. In this context, the optimization of culture conditions leading to the increase of the cytotoxicity of the bacterial broth was necessary to highlight bioactive compound(s).”

- (Lines 65 to 68): What is the reason for choosing B16 cell line to check the cytotoxicity? Authors should be explain more about the application such cytotoxic effect of broth extract on clarified targets such as therapeutic purposes.

The B16 cell line corresponds to murine melanoma cells. We used routinely these cells in our laboratory for screening cytotoxic activity of various extracts or compouds as B16 cells are a mouse model for human melanoma. 

To clarify the application of our work in therapeutic purpose the paragraph was modified as follows (p 4-5 lines 85-92): “In our attempts to focus on the discovery of novel agents to treat melanoma, one of the most fastest growing forms of cancer, we have selected the culture broth extract of this bacterium which showed an interesting cytotoxic activity against murine B16 melanoma cell lines (IC50 = 23 ± 3 µg/mL). In the previous work, the cytotoxic compounds involved in the biological activity of this strain were not identified. In this context, the optimization of culture conditions leading to the increase of the cytotoxicity of the bacterial broth was necessary to highlight bioactive compound(s).

- In addition, there is no explanation about the HaCaT cell line in introduction section of the manuscript.

The HaCaT cell line are the non-cancer cell line that we use routinely in our lab to evaluate the cytotoxic activity and to determine an eventual selectivity of the compounds tested.

It has been explained in the sentence p 5 lines 101-104: “At the end of the stationary growth phase the cultures were stopped and we determined the yield of extraction and the cytotoxic activity of the extracts against a cancer cell line (B16) and a non- cancer cell line (HaCaT) of the extracts which will eventually led us to determine a selectivity of action”

We have completed a sentence p 12 lines 262-263 as : Similar activities have been shown on HaCaT cells mentioning no selectivity of action.

- (Lines 69 to 70): Four parameters were targeted to evaluation in this study such as pH, Temperature, stirring speed and inoculum ratio, so please explain that based on which experimental design basically you reached to 81 experiments then in continue you reduced the number to 9 experiment?

The number of 81 experiments is necessary if you not use any experimental design. It corresponds to the number of trials necessary to test all the possible combinations with 3 levels of 4 parameters. The application of the Taguchi orthogonal design allow us to realise only 9 experiments to evaluate the impact of each parameter on the cytotoxic activity of the culture broth. It has been clarified by modification of the sentence p 5 lines 97-101: “The evaluation of all the possible combinations of these parameters requires 81 experiments, which is very time consuming. A Taguchi L9 orthogonal array design, which is a robust statistical DOE (Design of Experiments) method, allowing to study a set of variables with a limited number of trials, was employed to perform 9 experiments instead of 81 [17]”

4. Materials and method:

- (Line 110): How you ensure that the number of inoculated bacteria (precultured) were same in all experimental conditions? Please provide method for inoculum preparation and the number of bacteria (CFU/ml) added in both small scale and scaled up experimental condition.

All the precultures were obtained in a same way, the number of inoculated bacteria was controlled by monitoring the OD. It has been added in the manuscript p 6 lines 127-131 as: 

“Precultures

The precultures were obtained by inoculation of 30 mL of medium with an isolated colony collected on an agar plate. The culture was incubated until the optical density reached 0.6 for further uses. A purity control was performed on agar plate to check the absence of contamination.

- (Line 147): To keep the oxygen on 40% in the bioreactor, the cascade mode with the stirring was selected. Otherwise, based on experimental design the rpm is one of the variables which was adjusted on 150, 200 and 250 in each experimental condition. So the question arises is that when the amount of oxygen is reduced, the rpm may increase to compensate the aeration/oxygenation. Please explain how you justify such rpm variations during the experimental design.

The oxygenation reduced below 40% only during the exponential growth phase of the bacteria. As the production of compounds involved in the cytotoxic activity mainly happens during the stationary phase it means that at this time the stirring conditions are not affected by the cascade mode. The experimental design is based on variations occurred during stationary phase, to study the impact on the production of secondary metabolites.

- Please add the software name and version used for experimental design and statistical analysis. 

A sentence was added p 9 lines 206-607 : The experimental design and ANOVA analysis were performed using Excel 2016.

5. Results:

- The quality of figure no. 2 is not good. I suggest having alternate presentation of this figure.

The figure 2 has been retreated to increase the quality as suggested by the reviewer.

- The number of repetition in validation of the optimal condition and the results of them should be reported clearly in results section of the manuscript.

It has been modified p 17 lines 347-350 as: “The experiment B3 with the following parameters, pH 6, temperature 37°C, inoculum 5% and stirring 250 rpm, was repeated once using bioreactor to confirm the previous results on HaCaT and B16 cells.”

- The value of “yield” as a parameter should be followed by standard deviation.

The experimental design allowed to achieve the different culture conditions only once which explain why we do not have standard deviation for these results. It is the interest to do a experimental design. Only the optimal conditions were repeated. 

6. Discussion:

- According below statement of the manuscript, the differences between bioreactor and shake flask are well known and they are not much comparable to each other. I strongly recommend to revise the discussion part of manuscript.

“In fact, the stirring in bioreactor is performed by agitation blade submerged in the culture broth which implicates hyphal breaks, while in shaken flasks the agitation is performed with an orbital shaker allowing the hyphal formation. In a similar way, the aeration system in bioreactor consisted of a tube submerged in the culture broth that conducted oxygen into the medium causing the formation of bubbles, while in the shaken flasks there is no specific system for aeration. These main differences could explain the variation of bacterial production profile due to a different oxygenation and stirring methods inducing different stress conditions for bacteria.”

Following the reviewer’s comment, this part has been modified p 22 lines 461-462 as: “These variations of production could be explained by the different aeration and the stirring system used [28] leading to different stress conditions for bacteria.”

7. Minor comments:

- Some minor comment and suggestions are included in the pdf file of manuscript.

We have taken account all the comments of the reviewer 1. 

Reviewer #2: The work is interesting since explore the culture conditions to have the highest cytotoxic activity from Nocardia sp in bioreactor and flask. Cultures in flask result better since possibly cells are not as affected by hydrodynamic stresses as in bioreactor. The main different peaks separated by HPLC are not the responsible for the cytotoxic activity, so, there is pending to identify the molecules responsible. There have been identified some molecules with cytotoxic activity from Nocardia, so authors can trying to identify if activity could correspond to this kind of molecules.

---

## [Decision Letter · Decision Letter 1]

31 Dec 2019

Optimization of cytotoxic activity of Nocardia sp culture broths using a design of experiments

PONE-D-19-18890R1

Dear Dr. Tomasi,

We are pleased to inform you that your manuscript has been judged scientifically suitable for publication and will be formally accepted for publication once it complies with all outstanding technical requirements.

With kind regards,

Yoshihiro Uesawa

Academic Editor

PLOS ONE

Additional Editor Comments (optional):

Reviewers' comments:

Reviewer's Responses to Questions

**Comments to the Author**

1. If the authors have adequately addressed your comments raised in a previous round of review and you feel that this manuscript is now acceptable for publication, you may indicate that here to bypass the “Comments to the Author” section, enter your conflict of interest statement in the “Confidential to Editor” section, and submit your "Accept" recommendation.

Reviewer #1: All comments have been addressed

2. Is the manuscript technically sound, and do the data support the conclusions?

Reviewer #1: Yes

3. Has the statistical analysis been performed appropriately and rigorously? 

Reviewer #1: Yes

4. Have the authors made all data underlying the findings in their manuscript fully available?

Reviewer #1: Yes

5. Is the manuscript presented in an intelligible fashion and written in standard English?

Reviewer #1: Yes

6. Review Comments to the Author

Reviewer #1: (No Response)

7. PLOS authors have the option to publish the peer review history of their article (what does this mean?). If published, this will include your full peer review and any attached files.

Reviewer #1: Yes: Shayan Maleknia

---

## [Editor Report · Acceptance letter]

6 Jan 2020

PONE-D-19-18890R1 

Optimization of cytotoxic activity of *Nocardia* sp culture broths using a design of experiments 

Dear Dr. Tomasi:

I am pleased to inform you that your manuscript has been deemed suitable for publication in PLOS ONE. Congratulations! Your manuscript is now with our production department. 

With kind regards,

on behalf of

Dr. Yoshihiro Uesawa 

Academic Editor

PLOS ONE